# Role of Plasminogen Activation System in Platelet Pathophysiology: Emerging Concepts for Translational Applications

**DOI:** 10.3390/ijms23116065

**Published:** 2022-05-28

**Authors:** Filomena Napolitano, Nunzia Montuori

**Affiliations:** Department of Translational Medical Sciences and Center for Basic and Clinical Immunology Research (CISI), University of Naples Federico II, 80135 Naples, Italy; nmontuor@unina.it

**Keywords:** fibrinolysis, plasminogen, plasmin, plasminogen activation system, platelets, urokinase receptor

## Abstract

Traditionally, platelets have been exclusively considered for their procoagulant and antifibrinolytic effects during normal activation of hemostasis. Effectively, activated platelets secrete coagulation factors, expose phosphatidylserine, and promote thrombin and fibrin production. In addition to procoagulant activities, platelets confer resistance of thrombi to fibrinolysis by inducing clot retraction of the fibrin network and release of huge amounts of plasminogen activator inhibitor-1, which is the major physiologic inhibitor of the fibrinolytic cascade. However, the discovery of multiple relations with the fibrinolytic system, also termed Plasminogen Activation System (PAS), has introduced new perspectives on the platelet role in fibrinolysis. Indeed, the activated membrane surface of platelets provides binding sites on which fibrinolytic enzymes can be activated. This review discusses the evidence of the profibrinolytic properties of platelets through the description of PAS components and related proteins that are contained in or bind to platelets. Our analyses of literature data lead to the conclusion that in the initial phase of the hemostatic process, antifibrinolytic effects prevail over profibrinolytic activity, but at later stages, platelets might enhance fibrinolysis through the engagement of PAS components. A better understanding of spatial and temporal characteristics of platelet-mediated fibrinolysis during normal hemostasis could improve therapeutic options for bleeding and thrombotic disorders.

## 1. Background

### 1.1. The Plasminogen Activation System and Its Regulation

Hemostasis is the process that blocks blood loss from a damaged vessel through the polymerization of fibrin monomers and entrapment of blood cells, thereby generating the blood clot [1]. Fibrinolysis is the process that leads to clot dissolution by hydrolysis of fibrin, which derives from the coagulation cascade following cleavage of fibrinogen by thrombin, and it is catalyzed by the Plasminogen Activation System (PAS) [2]. In this way, the balance between coagulation and fibrinolysis promotes the normal hemostasis.

PAS is a complex proteolytic cascade reaction that regulates the formation and activity of plasmin, a broad-spectrum serine protease. To perform its multiple functions, PAS is regulated by an array of activators, inhibitors, and cellular receptors that are briefly reviewed below.

The central component of the fibrinolytic cascade is plasminogen, which is synthesized by the liver and released as zymogen into the circulation at a concentration of 2 μM. It is composed of five kringle-like domains containing “lysine-binding sites” and a C-terminal domain homologous to other trypsin-like proteases. The native form of circulating plasminogen has glutamic acid (Glu) at the N-terminal and is called Glu-plasminogen. Other plasminogen forms generated by the catalytic cleavage of plasmin and containing mostly lysine (Lys) at the N-terminal position are indicated as Lys-plasminogen. After cleavage of the Arg^561^–Val^562^ specific bond by plasminogen activators, single-chain plasminogen is transformed into two-chain plasmin, in which the two-chain (heavy A and light B chains) are connected by a disulfide bond. The active site of plasmin is located in the light B chain and corresponds to the amino acid triad of Ser^741^, His^603^, and Asp^646^ [3,4,5].

The reaction of conversion of the inactive plasminogen to plasmin can be catalyzed by two immunologically distinct plasminogen activators: tissue-type plasminogen activator (tPA) and urokinase plasminogen activator (uPA).

The serine proteinase tPA is synthesized and released in a single-chain form, which can be converted to a two-chain form by proteolytic cleavage of a single polypeptide bond (Arg^275^-Ile^276^). The two chains are held together by a disulfide bond. The plasminogen activation activity of single-chain tPA is 10- to 50-fold lower than that of the two-chain form. Both forms of tPA play an essential role in fibrinolysis, as it possesses a high affinity to fibrin, similar to plasminogen [6,7].

The serine proteinase uPA, encoded by the PLAU gene, is minimal in normal cells, whereas in cancer cells, the expression is significantly increased. uPA was first identified in urine, but further study demonstrated its presence in other biological fluids such as serum, plasma, or seminal fluid [8]. uPA possesses no affinity to fibrin and activates plasminogen primarily within physiological and pathological tissue remodeling processes [9]. uPA is synthesized and released as a single polypeptide chain glycosylated zymogen termed sc-uPA or also named pro-urokinase (pro-uPA) that undergoes cleavage of the peptide bond between Lys^158^ and Ile^159^ located at a linker region to produce a two-chain form uPA linked via a disulfide bond [10]. It has been shown that pro-uPA is 250-fold less potent in generating plasmin than the two-chain uPA [11]. Among different proteases able to mediate the cleavage of pro-uPA, plasmin is the most effective. uPA binds to the cell surface through the high-affinity uPA receptor (uPAR) that mediates, in addition to plasminogen activation, a variety of biological activities at the cell surface, such as extracellular matrix (ECM) remodeling, cell adhesion, cell migration, and intracellular signaling. uPAR consists of a single polypeptide chain cysteine-rich glycoprotein with three distinct domains (DI, DII, and DIII) connected to the cell membrane by a glycosylphosphatidylinositol (GPI) anchor [12]. The DI–DII linker region is particularly sensitive to the activity of several proteases, such as trypsin, chymotrypsin, elastase, metalloproteases, and uPA itself. The cleavage in the DI–DII linker region causes DI release and the generation of truncated forms of uPAR (DIIDIII–uPAR) on the cell surface. Both full-length and cleaved forms of uPAR can be subjected to the cleavage of the GPI anchor. This further cut generates soluble forms of uPAR that circulate in the blood at a very low level, but its circulatory level is elevated in cancer and in diverse inflammatory conditions [12,13].

Fibrin itself is an active promoter of plasminogen conversion to the active form, providing a surface for both plasminogen and tPA binding, thus accelerating tPA-mediated cleavage of plasminogen into plasmin. The presentation of exposed C-terminal lysine residues in fibrin allows plasminogen to position itself through its lysine binding sites and to be more efficiently cleaved by tPA [14].

The activity of plasmin is negatively modulated at different levels in order to prevent hyperfibrinolytic conditions. At the level of plasminogen activators, plasminogen activator inhibitor type-1 (PAI-1) and type-2 (PAI-2), belonging to the serpin family, cause the inhibition of uPA and tPA [15]. Regulation of plasmin activity is performed by the plasma proteinase inhibitors, α2-antiplasmin and α2-macroglobulin [7]. At the level of the blood clot, thrombin-activatable fibrinolysis inhibitor (TAFI) removes C-terminal lysine residues on partially degraded fibrin in order to lead to decreased plasmin formation and thus stabilization of the clot [16].

Great advances have been made in the understanding of the plasminogen activation system, from the initial discovery of proteolysis of fibrin by plasmin to the multifaceted role of this system. In addition to fibrinolysis, PAS plays a central role in multiple physiological and pathological processes, including ECM degradation, embryogenesis, cell migration, tissue repair, wound healing, angiogenesis, inflammation, tumorigenesis, and metastatization [8,17,18,19,20].

Several studies point to the importance of fibrin-independent functions of plasmin during acute inflammation and inflammation resolution. More recently, Napolitano and Montuori have elucidated the ability of plasmin and components of the PAS to contact other molecular pathways and to influence many pathophysiologic processes. In fact, the PA system, located at the interface between the kallikrein pathway and the complement system, is involved in a positive feedback reaction from which pro-inflammatory protein fragments or peptides are released [21].

Most of the available data on the pathophysiology of PAS concern different cell types such as epithelial, mesenchymal, cancer, stem, and immune cells. Although fibrinolysis and coagulation are two closely related processes, platelets have been extensively characterized for their role in coagulation but not in the fibrinolytic cascade.

### 1.2. The Complexity and the Diversity of Platelets

Platelets are anucleate cytoplasmic fragments derived from the megakaryocytic precursor and circulate in the blood for 5–7 days [22]. Platelets were traditionally considered crucial to the hemostasis and thrombosis processes, providing the ability to arrest bleeding. They are complex cells that contain three distinct types of secretory compartments: dense granules, α-granules, and lysosomes, through which they store and release bioactive mediators involved in several pathophysiological processes [23]. The dense granule components, such as polyphosphate and small molecules such as ADP, ATP, Ca^(2+)^, and serotonin, play a central role in platelet aggregation to form the hemostatic plug. The α-granules are the most abundant (approximately 50–80 granules per platelet) and can release constituents with opposing activities such as a prothrombotic and antithrombotic activity or angiogenic and antiangiogenic activities [24]. In fact, α-granules contain chemokines, proteoglycan, Willebrand factor (VWF), fibrinogen, P-selectin, and complement binding proteins [25]. Lysosomes contain hydrolases able to eliminate the circulating platelet aggregate [26].

Upon tissue trauma, platelets adhere to the ECM through different platelet receptors that bind matrix adhesive macromolecules. Platelet adhesion to ECM leads to tethering and rolling over the damaged vessel wall and to a signaling cascade mediated by tyrosine kinases and G-protein-coupled receptors, which implies full activation of platelets and granule release, in turn resulting in recruitment of additional platelets. Importantly, ADP is released from dense granules when platelets adhere to the sites of vascular injury and is primarily responsible for platelet activation, recruitment, and induction of aggregation. After the initial platelet plug, engagement of the coagulation factors via platelet exposure of phosphatidylserine (PS) leads to fibrin mesh formation that enhances the thrombus [27]. Indeed, platelets themselves are an important source of coagulation factors such as Factors V and XIII, polyphosphates, and prothrombin [28].

Until a few years ago, the roles of platelets were considered limited to primary hemostasis with initial aggregation and plug formation. To date, the importance of platelets has been recognized in each of the events leading to vessel repair, including immune cell recruitment, inflammation, wound healing, angiogenesis, and tissue remodeling [29]. Probably, the ability of the platelet to release multiple mediators from the intracellular compartment (also called “secretome” or “sheddome”) is a major driver of these functions, but the mechanisms that regulate and coordinate the spatiotemporal release of intracellular content are still unknown [30].

Further, it has been demonstrated that platelet activation following vascular injury is heterogeneous in receptor and signaling proteins [31]. Different populations of platelets with distinct surface patterns have been identified: i) aggregating platelet binding fibrinogen via α_IIb_β_3_ integrin (glycoprotein IIb/IIIa; GPIIb/IIIa) and exposing FXIII-A; ii) procoagulant coated platelets exposing phosphatidylserine (PS) and binding coagulation factors; iii) contracting platelets with cell–cell contacts [32]. The recognition of platelet heterogeneity within the thrombus has introduced new insights into the role of platelets as sentinels of vascular integrity.

### 1.3. Platelets as Balance between Thrombus Formation and Fibrinolysis: An Emerging Concept

Platelets have long been considered for their role in promoting thrombus resistance to fibrinolysis. Classically, the principal mechanism through which platelets induce fibrinolytic resistance is clot retraction, also named clot contraction. This process is triggered by fibrin binding to platelet α_IIb_β_3_ integrin and is aimed at generating contractile forces that change the structure of the fibrin network and reduce clot permeability. The final step is the formation of dense fibrin networks resistant to fibrinolysis due to condensing of α2-antiplasmin cross-linked to fibrin and decreased binding of tPA [33,34]. In platelet-rich clots, the availability of plasminogen and plasminogen activators is attenuated by clot retraction due to the extrusion of these proteins [35].

In addition to clot retraction, platelets exert antifibrinolytic properties through other two relevant pathways: (i) platelets are a focal point of fibrin formation because of their ability to facilitate thrombin generation and to directly interact with circulating fibrinogen [28]; (ii) platelets secrete fibrinolytic inhibitors, such as α2-antiplasmin, PAI-1, and TAFI, which are all contained at high concentrations within α-granules [36].

Nevertheless, recent developments in the field indicate that the role of platelets in fibrinolysis is contradictory: on the one hand, platelets confer resistance to lysis of fibrin networks by inducing clot retraction and PAI-1 release; on the other hand, platelet surface constitutes a binding site for fibrinolytic proteins. In particular, the activators of the plasminogen cascade can either be released following platelet degranulation or bind to specific platelet receptors. This implies that the fibrinolytic process could be regulated by platelets in order to remove the hemostatic plug after the healing process.

It seems that platelets play a controversial role, but most likely, platelet functions are finely regulated in space and temporal manner for the entire hemostatic process. This tight regulation would allow the platelets to adapt to the different phases of hemostasis, thus explaining the switch from antifibrinolytic to profibrinolytic profile.

## 2. Molecular Connections between Platelets and Components of Plasminogen Activation System

This section focuses on the main scientific evidence of the relationship between the components of PAS and platelets, its effects on physiological hemostasis, and its contribution to pathological conditions.

### 2.1. Plasminogen as Zymogen Form

The possibility that plasminogen is released from platelet organelles or that platelets express specific receptors for plasminogen binding outlines a new chapter on the platelet functions in the physiological and pathological hemostasis. The increasingly accepted hypothesis is that platelets provide a mechanism for the localization of plasminogen at a site of fibrin formation, thereby enhancing the fibrinolytic cascade [37].

Firstly, Coppinger et al. showed that platelets are able to release plasminogen upon thrombin stimulation [38]. Proteomic analysis of platelet α-granules using mass spectrometry demonstrated that plasminogen is contained in α-granules, but it has not yet clarified whether the alpha-granular pool contains Lys- or Glu-plasminogen [35]. Notably, plasminogen was undetectable in megakaryocyte cultures unless the cell culture was exposed to the media containing plasma, probably due to an endocytic mechanism by these cells [39]. Secondly, it has been established that plasminogen interacts with human platelets, and the binding is enhanced fivefold by thrombin stimulation [40]. Multiple receptors for plasminogen have been described in various cell types, and a common feature of these receptors is that they interact with plasminogen via its lysine binding sites located in the kringle domains. This lysine-dependent binding mechanism was confirmed by the treatments with lysine analog ε-aminocaproic acid and carboxypeptidase B [40,41,42]. It has been established that plasminogen, associated with the platelet surface, assumes an open conformation that is more readily cleaved to plasmin by activators and is protected from inhibition by α2-antiplasmin [40,43,44,45].

Of the platelet surface proteins, the integrin α_IIb_β_3_ seems to be the main plasminogen-binding site. Miles and Plow demonstrated that plasminogen binds to the fibrinogen receptor α_IIb_β_3_ on platelets and that thrombin stimulation increases plasminogen binding [40]. Subsequently, Adelman et al. showed that plasminogen binding to platelet α_IIb_β_3_ may be mediated via plasminogen associated with fibrinogen through lysine binding domains [42].

Whyte et al. provided new insights on the location and the role of plasminogen in in vitro model for thrombus formation and lysis under flow conditions. The authors found that adherent spread platelets accumulate plasminogen in an α_IIb_β_3_-dependent manner, whereas PS-exposing platelets directly bind plasminogen through protruding caps or indirectly via platelet-associated fibrinogen. Notably, PS-exposing platelets did not express α_IIb_β_3_, as demonstrated by the lack of PAC-1 (activated GP IIb/IIIa) staining in the platelet caps. Altogether, these data indicated that PS-exposing platelets foster the conversion of plasminogen into plasmin, thus modulating local fibrinolysis within the microenvironment of the thrombus [46].

Plg-R_kt_ is an integral transmembrane receptor synthesized with a C-terminal lysine, firstly identified on the macrophage surface, which promotes plasminogen activation, cell migration, and polarization. More recently, Whyte et al. demonstrated the presence of Plg-R_kt_ on the platelet surface. Genetic ablation of Plg-R_kt_ reduces plasminogen binding by roughly 50%, suggesting the central role of this receptor in the retention of plasminogen on the activated platelets [47].

### 2.2. Plasmin

It is conceivable that platelets are involved in the plasmin positive feedback loop; once released from platelets, plasmin can participate in mechanisms that further enhance platelet activities. However, it is still unclear whether plasmin exerts an activating or inhibiting action on platelets [48]. It is certain that experimental temperature, incubation time, plasmin concentration, and presence of plasma proteins modify plasmin effects on human platelets in vitro [37].

Niewiarowski et al. were the first to introduce the concept of plasmin as a platelet activator. The authors demonstrated that plasmin could induce reversible platelet aggregation and significant granule release. The short treatment of platelets with plasmin enhanced their sensitivity to ADP, a key agonist for platelet aggregation [49]. The study of biochemical mechanisms indicated that plasmin at high doses caused aggregation and secretion of washed human platelets associated with Ca^2+^ mobilization and phospholipase C and protein kinase C activation, suggesting the involvement of G_q_ protein [43]. In line with these studies, Ishii-Watabe et al. suggested that plasmin directly induced platelet aggregation via ADP secretion from platelets. In addition, the authors showed that platelet shape change occurred just after the stimulation with plasmin, while several minutes were required for granule release and aggregation [50]. Interestingly, platelet incubation with lower plasmin concentrations for 5 min at room temperature enhanced platelet aggregation, integrin α_IIb_β_3_ expression, α-, δ-granules, and lysosomes release [51,52]. The data obtained from Quinton et al. indicated that plasmin induced platelet aggregation primarily through cleavage of Protease-Activated Receptors (PAR)-4 [51].

It could be of clinical relevance to investigate the molecular aspects of the regulation of plasmin-mediated platelet activation. More recently, Pielsticker et al. indicated plasmin as a potent platelet activator, inducing P-selectin expression and fibrinogen binding. Moreover, the authors identified matricellular glycoprotein thrombospondin-1 (TSP-1) as a promoter or an inhibitor of plasmin-induced platelet activation, depending on its concentration, when bound to plasmin [53].

An important factor influencing the platelet response to plasmin in vitro is the platelet environment. Blockmans et al. reported the aggregatory effects of plasmin on gel-filtered platelets but no aggregation response in PRP, probably due to the presence of α2-antiplasmin in PRP [54]. In contrast to these data, Pielsticker et al. observed plasmin activating effects on platelets not only in the absence of plasma factors but also in diluted platelet-rich plasma and citrated whole blood, despite the presence of the plasmatic plasmin inhibitors α2-antiplasmin and α2-macroglobulin [53].

Related to the inhibitory effects of plasmin on platelets, Schafer and Adelman found that plasmin inhibited arachidonic acid mobilization from platelet membrane phospholipid pools [55]. Plasmin treatment of washed platelets resulted in progressive loss of GPIb, which contains receptor sites for von Willebrand (vWF), thereby reducing vWF-mediated platelet agglutination [48]. In the presence of plasma, the incubation of human platelets with streptokinase, able to produce plasmin, resulted in impairment of aggregation [56].

In conclusion, plasmin seems to exert contrasting effects on platelets, being able to both activate and inhibit platelet activity in vitro. This bivalent behavior could be caused by several variables, such as temperature and/or incubation time, that can modify and influence the platelet response to plasmin in vitro, but a consistent body of evidence predominantly indicates that plasmin acts as a platelet activator.

### 2.3. Tissue-Type Plasminogen Activator

The potential role for platelets in the modulation of fibrinolysis became apparent when Vaughan et al. demonstrated that the platelet surface possessed for tPA a large number of specific, low-affinity binding sites that were amplified upon thrombin stimulation [57]. However, specific receptors for tPA on the platelet surface have not yet been identified. In addition, the presence of tPA has been detected by different techniques in isolated megakaryocytes, as well as in platelets [58,59].

tPA released from endothelium protects against the formation of platelet aggregates in vivo by modulation of platelet cyclic nucleotides [60].

The thrombolytic activity of tPA was impaired by platelets in vitro as a consequence of clot retraction and decreased binding of the plasminogen activator [33]. This may explain the resistance of platelet-rich clots to lysis in vivo.

Importantly, the “lisability” of platelet-rich areas depends on the complex interrelationships between the mechanical platelet contractile forces and the process of fibrinolysis. In fact, GP IIb/IIIa inhibitors facilitated the rate and the extent of fibrinolysis by improving recombinant tPA binding velocity and, subsequently, the lysis rate in platelet-rich areas, thus providing new insights into the synergistic potential of GP IIb/IIIa inhibitors and fibrinolytic agents in dissolving thrombotic vaso-occlusions [61].

A model of platelet lysate affinity chromatography showed that a number of platelet proteins that have previously been found to inhibit the fibrinolytic system were captured by tPA. Since there is evidence that platelets undergo lysis after trauma, the authors hypothesized that platelet lysis could shut down tPA-mediated fibrinolysis, with implications for therapeutic intervention, in order to balance fibrinolysis during resuscitation of trauma patients [62].

### 2.4. Urokinase Plasminogen Activator

Plasminogen activator uPA plays a central role outside the vasculature by interacting with its receptor uPAR on the cell surface, but several studies reported an active role also in fibrinolysis.

Platelets from normal subjects contain a small amount of uPA (up to 1.3 ng/10^9^), while insignificant levels were found in megakaryocytes [63]. It was shown that about 20% of pro-uPA intrinsic to blood was closely related to platelets and that platelets were able to rapidly incorporate exogenous pro-uPA [64].

pro-uPA seems to be involved in a mechanism for stimulation of fibrinolysis by platelets. In fact, it was observed that pro-uPA could mediate the clot lysis, and this mechanism was enhanced by platelets. Loza JP et al. reported that platelet-bound prekallikrein was able to enhance pro-uPA-induced clot lysis [65], but most probably, platelet-bound plasminogen is the essential player in this phenomenon. In fact, platelets from plasminogen-deficient mice were not able to activate pro-uPA, and the activation of pro-uPA was significantly enhanced by plasminogen on the platelet surface rather than plasminogen in the solution phase [44]. Interestingly, fragment E-2 from fibrin promoted plasminogen activation by pro-uPA, and fibrin specificity to pro-uPA didn’t require its conversion to uPA [66]. Furthermore, in the presence of plasma, activation of Glu-plasminogen bound to degraded fibrin was found to be more efficient with pro-uPA than with uPA [67]. In plasma, the pro-uPA fibrinolytic property was stable for several days and, unlike uPA, did not form stable inhibitor complexes [68]. Altogether, these findings suggest the use of pro-uPA as a thrombolytic agent against platelet-rich thrombi, which are more resistant to lysis.

Storage of large amounts of uPA in blood platelets was observed in a congenital bleeding disorder known as Quebec Platelet Disorder (QPD). Unlike normal platelets, QPD platelets contain approximately 400–600 ng/10^9^ of uPA without increased uPA in plasma or systemic fibrinolysis [69]. uPA from QPD platelets is predominantly detected as active two-chain uPA and in smaller amounts as single-chain and/or forms complexed to PAI-1. The high levels of active uPA were accompanied by abnormal intraplatelet generation of plasmin. During clot formation, uPA released by QPD platelets led to platelet-dependent increased fibrinolysis, thus provoking QPD bleeding. QPD platelets showed the loss of normal α-granule proteins, and this may possibly contribute to QPD bleeding. The genetic defect that causes QPD is characterized by a duplication mutation that selectively induces increased production of normal PLAU transcripts by megakaryocytes [70].

Another mechanism for uPA-mediated plasmin formation has been reported: glu-plasminogen bound to platelet surface was converted to plasmin with a high efficiency by uPA expressed on monocytes or endothelial cells-derived microparticles [45]. Importantly, uPA associated with platelet surface was able to upregulate uPA mRNA synthesis by endothelial cells [71]. Through this mechanism, platelets may control intravascular fibrin deposition.

The connection between platelet functions and plasminogen activator uPA has also been found in angiogenesis. It is known that platelets are an important circulating store of angiogenesis regulators, such as the angiogenesis inhibitor angiostatin. Jurasz et al. showed that angiostatin generation by platelets was inhibited by a selective inhibitor of uPA. The authors concluded that platelets constitutively generate angiostatin on their membranes, and this mechanism is mediated by uPA [72].

### 2.5. High-Affinity uPA Receptor (uPAR)

It has been established that uPA receptor (uPAR) is expressed on the platelet surface, but data on uPAR involvement in platelet pathophysiology are not available. However, a critical role of platelets uPAR for kinetics and endothelium adhesion associated with inflammation has been demonstrated [73]. Sloand et al. reported increased suPAR levels in the sera of patients affected by paroxysmal nocturnal hemoglobinuria (PNH). In PNH patients, GPI-negative granulocytes and platelets were the probable source of elevated plasma suPAR levels that were associated with thrombosis and inhibition of plasmin generation [74].

The cleaved form of uPAR (DII-DIII-uPAR) containing the SRSRY sequence corresponding to amino acids 88–92 at its N-terminus interacts with N-formyl peptide receptors (FPRs), thus mediating cell migration [75]. FPRs belong to a family of G protein-coupled receptors and are involved in the regulation of innate immunity and host defense. Three family members are identified: FPR1, FPR2/ALX, and FPR3, mainly expressed in several types of innate immune cells, including neutrophils and monocytes/macrophages; other cell types also express FPR members. Recently, our group assigned an important pathogenetic role to crosstalk between uPAR and FPRs in inflammation and cancer [75,76]. Despite numerous research focused on the role of FPRs in the modulation of immune responses, little data on their ability to regulate hemostasis and thrombosis are available. Czapiga M et al. demonstrated the expression of FPR1 on the platelet surface and its ability to induce chemotactic and migratory functions upon stimulation with E coli-derived fMLF [77]. Salamah et al. showed the expression of FPR2 and its effects in promoting platelet activation and thrombus formation under arterial flow conditions [78].

Further studies are necessary to establish the role of FPRs on platelets and their contribution to hemostasis and thrombosis, but it would be interesting to investigate whether crosstalk between uPAR and FPRs occurs on the platelet surface and whether it is involved in the regulation of FPRs-mediated effects on platelets.

### 2.6. Plasminogen Activator Inhibitors (PAI-1 and PAI-2)

The main endogenous inhibitor of plasminogen activators and the most important modulator of the fibrinolytic system is PAI-1. Several cell types are capable of synthesizing, storing, and releasing PAI-1, but platelets are considered a major reservoir of plasma PAI-1 [79]. However, there is an ongoing scientific debate on the effective contribution of platelet PAI-1 to fibrinolysis.

About 90% of the total PAI-1 in blood derived from platelet α-granules and platelet count were correlated with plasma PAI-1 concentrations [80]. Recent findings showed that more than 50% of platelet PAI-1 is in the active configuration, contrary to preliminary studies where the majority of PAI-1 stored in platelets has been considered to be inactive [81]. Because platelets retain mRNA from megakaryocytes and a capacity for synthesis of some proteins, platelets can de novo synthesize PAI-1, and the amount synthesized in vitro in 24 h is 35-fold higher than required to maintain normal plasma levels. Interestingly, Brogren et al. investigated the hypothesis that platelets might be the source of plasma PAI-1 and that the cellular source of PAI-1 can be determined by its tissue-specific glycosylation pattern [79]. PAI-1 isolated from macrophages, endothelial cells, and adipose tissue expressed heterogeneous glycosylation patterns, but no glycans were detected on PAI-1 isolated from plasma or platelets from healthy individuals, thus suggesting that platelets may be the main source of plasma PAI-1 [82].

The study of platelet PAI-1 provided multiple evidence that it plays a central role in fibrinolytic resistance to clots and thrombi. In vitro studies showed that platelet-mediated fibrinolytic resistance is reduced by neutralizing antibodies direct against PAI-1 [83]. In platelet-rich plasma clot lysis mediated by tPA, the release of PAI-1 from activated platelets resulted in a prolongation of the clot lysis time [84].

In order to elucidate the mechanisms responsible for the stability of platelet-rich thrombi, in vitro model thrombus (composed of thrombin-activated platelets, fibrin, plasminogen, and tPA) provided novel insights into the role of platelet PAI-1. In particular, PAI-1 released from activated platelets stabilizes fibrin deposited on the platelet surface and thus increases the lysis resistance of platelet-rich thrombi [83].

PAI-1 circulates in complex with vitronectin (VN), an interaction that stabilizes PAI-1 in the active form [85]. VN can be internalized into platelet α-granules from plasma, but its effects on the platelet pool of PAI-1 are not clear. Confocal microscopic analysis of platelet-rich plasma clots confirmed the co-localization of PAI-1 with VN and vimentin on the activated platelet surface, suggesting that platelet vimentin may regulate fibrinolysis by binding platelet-derived VN/PAI-1 complexes [86].

Animal model studies established that platelet PAI-1 was primarily involved in arterial thrombi, which contain high platelet content [37]. In particular, Zhu et al. noted that, in a murine model of carotid artery injury, the reperfusion after tPA treatment occurred in PAI-deficient mice but not in wild-type animals, thus demonstrating the main role of PAI-1 in the thrombolytic resistance of platelet-rich thrombi [87].

In line with in vitro studies and data from animal models, the immunohistochemical analysis of human thrombi suggested that active recruitment of platelets contributes to the high PAI-1 concentration in thrombi [88].

Diverse pathological conditions are associated with unregulated PAI-1 levels, but only plasma PAI-1 levels in most cases were examined.

Platelets were capable of immediate degranulation when exposed to thrombin leading to the release of active PAI-1, which is complexed with tPA and inhibits tPA-induced fibrinolysis [89]. Following degranulation, platelets synthesize more PAI-1, but such synthesis occurs over a long period of time, rendering these platelets dysfunctional. Since thrombin levels are elevated in trauma and intensive care unit (ICU) patients, these observations deserve more attention.

Platelet PAI-1 could play a central role in diverse pathological conditions associated with hypoxia, such as chronic obstructive pulmonary disease (COPD). In fact, exposure of human platelets to hypoxia enhanced PAI-1 expression. In addition, circulating platelets isolated from COPD patients had higher PAI-1 levels compared to controls, and probably, the prothrombotic phenotype associated with hypoxia may be attributable to the synthesis of PAI-1 by platelets [90].

It has been proposed that miRNA plays a central role in the regulation of fibrinolysis [91]. Platelets of patients with type 2 mellitus diabetes (DM2) release significantly more PAI-1 at the same level of platelet aggregation. This may contribute to enhanced thrombosis in diabetes [92]. Moreover, recent studies showed that mir-30c negatively regulated PAI-1 mRNA and protein expression in megakaryocytes; hyperglicemia-induced repression of mir-30c enhanced PAI-1 expression and thrombus formation in DM2 [93].

In congenital bleeding disorders, low levels of intraplatelet PAI-1 were found, but it is difficult to establish the relationship between deficiency of platelet PAI-1 and bleeding [37].

A significant increase in platelet PAI-1 levels was observed in essential thrombocythemia (ET) patients with thrombotic complications compared to ET patients without thrombotic complications and control group [94].

Taken together, the evidence strongly suggests that platelet PAI-1 plays a central role in arterial thrombosis and in hemostasis, as it is a determinant factor for fibrinolytic resistance.

PAI-2 is classically considered as an inhibitor of fibrinolysis via inhibition of plasmin generation by uPA and, to a lesser extent, tPA. Novel functions were attributed to PAI-2, which seems to be involved in hemostasis and to be associated with platelets. In particular, PAI-2 deficiency in mice reduced bleeding times via dysregulated platelet activation. The immunohistochemistry analysis of blood clots demonstrated that PAI-2 was associated with platelet aggregates; however, PAI-2 was not detected in human platelets [95].

### 2.7. Alpha 2-Antiplasmin

α2-antiplasmin is the principal inhibitor of plasmin and can be detected in platelet α-granules. α2-antiplasmin antigen contained in platelets corresponded to 0.05% of the total α2-antiplasmin present in the blood on a volume basis, and 87.5% of this material can be released when platelets are stimulated by thrombin. Immunohistochemical analysis of human thrombi showed that α2-antiplasmin was present at high concentrations, but platelets released small amounts of α2-antiplasmin [37,88,96]. However, thrombus stabilization against fibrinolysis involves this mechanism: platelet FXIII-A through exposure to the activated platelet membrane exerts antifibrinolytic function by cross-linking α2-antiplasmin to fibrin [97].

Takei M et al. examined the effects of α2-antiplasmin on platelet aggregation using animal models and found that lack of α2-antiplasmin increased platelet micro-aggregation when platelets were stimulated with ADP. Microaggregates produced in the early phase of platelet activation have the potential to aggravate thrombus formation, leading to vascular occlusions by the formation of a platelet-rich thrombus [98].

## 3. Discussion

It is widely established that platelet-rich thrombi are more resistant than platelet-poor thrombi to both endogenous fibrinolysis and thrombolytic therapy. However, newly discovered pathways of platelets in hemostasis and thrombosis are changing the classic view of the platelets as major players in conferring fibrinolytic resistance to blood clots or thrombi. In this scenario, we highlight the possibility that platelets act as a delivery service for components of PAS, thus providing both antifibrinolytic and profibrinolytic potentials.

Platelets interact with fibrinolytic factors in a complex manner, and they serve as a focal site for the assembly of PAS (Figure 1). Plasminogen can either be released from platelet α-granules or be recognized by specific receptors expressed on the platelet surface. When bound to platelet surface, plasminogen assumes an open confirmation that it can be cleaved to plasmin by plasminogen activators. Plasminogen activators, uPA and tPA, can be provided by platelets, and in turn, they bind the platelet surface through specific receptors. While platelet receptors for tPA have not yet been identified, platelets express both membrane uPA receptor (uPAR) and release GPI-negative uPAR, also named soluble uPAR (suPAR). Both GPI-positive and GPI-negative uPAR can interact with FPRs expressed on the platelet surface; thereby, they could induce platelet activation, migration, and thrombus formation. A platelet profibrinolytic effect mediated by uPA is believed to be the main pathogenetic cause of a rare bleeding disorder known as Quebec platelet disorder (QPD).

Once activated, plasmin can exert both stimulatory and inhibitory effects on platelet: (i) plasmin activates platelets through PAR-4 receptors, inducing Ca^2+^ mobilization, phospholipase C, and protein kinase C activation; (ii) plasmin increases secretion from lysosomes; (iii) the treatment with plasmin induces ADP release from dense granules, thereby playing a crucial role in hemostasis and thrombosis; (iiii) plasmin induces α_IIb_β_3_ and P-selectine expression. The stimulation of α-granules released by plasmin could generate an auto-amplification loop through which plasminogen and uPA are released. Interestingly, thrombospondin-1, which is released from α-granules, depending on the concentration, could act as a cofactor and/or inhibitor of plasmin-induced platelet activation.

To counterbalance the activation processes of the fibrinolytic cascade, platelets are able to release inhibitors of PAS. Firstly, platelets are considered a major reservoir of plasma PAI-1, which is an inhibitor of plasminogen activators. Moreover, platelet α-granules can release α2-antiplasmin, the principal inhibitor of plasmin. α2-antiplasmin is cross-linked to fibrin by activating coagulation factor XIII (FXIII-A) in order to promote thrombus stabilization.

The activation of both platelets and the fibrinolytic cascade is tightly regulated in a space- and time-dependent manner. In the hemostatic process, platelets could have the potential to both inhibit and activate fibrinolysis. In the early phase after vascular injury, platelets accumulate the inhibitors of PAS at a site of the core of the thrombus and confer resistance to fibrinolysis by continuous afflux of PAI-1 and α2-antiplasmin. However, in vivo studies showed Glu-plasminogen binding to either the surface of activated platelets or to fibrin in the early phase of microthrombus formation, even though the size of the thrombus was unchanged [99]. These data support the hypothesis that the high levels of PAI-1 and cross-linked α2-antiplasmin does not allow thrombus lysis in the early phase. In time, the consumption of inhibitors of PAS, the continuous binding of plasminogen and plasmin generation on platelet surface can initiate the lytic phase of the thrombus, leading to a switch from antifibrinolytic to profibrinolytic phenotype.

Another important aspect is that platelets are heterogeneous cells that exist in different activated states and, thus, expose different receptors and molecules on their surface. A better understanding of the fibrinolytic profile of each platelet subpopulation is necessary to determine which of these is important for mediating thrombolysis.

Altogether, this knowledge can be applied in several clinical settings through the development of novel targeted therapies. The ability of PAS components to activate and inhibit, at the same time, platelet functions may be relevant during the treatment of thrombotic disorders. In fact, the activation of the fibrinolysis through current thrombolytic agents (such as recombinant tPA) can cause the platelet activation and, thus, contribute to reocclusion or late reperfusion, which are the major clinical complications associated with thrombolytic therapies. Moreover, the combined use of thrombolytic agents and antiplatelet drugs could be associated with an increased risk of bleeding or modification of hematologic parameters. In addition, several pieces of evidence indicate that platelet-rich thrombi, particularly arterial thrombi, are resistant to current therapeutic regimens because the active recruitment of platelets contributes to the high PAI-1 concentration in thrombi [87].

From our studies on the paradoxical relationship between PAS and platelets, it emerges that attention must be shifted from the effector components of the fibrinolytic system toward regulatory components, such as thrombospondin-1. Future studies should shed light on the contribution of thrombospondin-1 in platelet activation and thrombus formation; in fact, targeting the thrombospondin-1 pathway may represent a promising thrombolytic strategy for controlling platelet activation in thrombotic diseases.

Interestingly, the pharmacological inhibition of FPRs/uPAR crosstalk on the platelet surface could be a starting point for the development of preventive therapies against thrombus formation.

The other important aspect is the management of bleeding disorders, common conditions that can be caused by congenital or acquired diseases. Antifibrinolytic agents, primarily tranexamic acid, are used to reduce bleeding and blood transfusion, being able to prevent directly or indirectly the activation of plasminogen to plasmin. The growing understanding of the molecular mechanisms through which plasmin exerts its effects could help develop new targeted therapies. For example, monoclonal antibodies or small molecules directed against Plg-R_kt_ could compete with the plasminogen binding to the cell surface, or else the neutralization of tPA or uPA activities could prevent plasminogen activation.

## 4. Conclusions

Despite their wide reputation as procoagulant and antifibrinolytic agents, platelets remain a surprising cell type in which novel functions continue to be discovered.

Starting from literature data that demonstrate the ability of platelet to release plasminogen activators and to bind plasminogen, we discuss new emerging concepts on the role of platelets during the hemostatic process. In normal hemostasis, platelets’ antifibrinolytic potential overcomes profibrinolytic activity in the early phase of thrombus formation, but this condition is reversed by the subsequent lytic phase. Better knowledge of platelet-mediated modulation of fibrinolysis and its spatiotemporal regulation could have important implications for developing or improving novel thrombolytic treatments for different disease conditions.

## Figures and Tables

**Figure 1 ijms-23-06065-f001:**
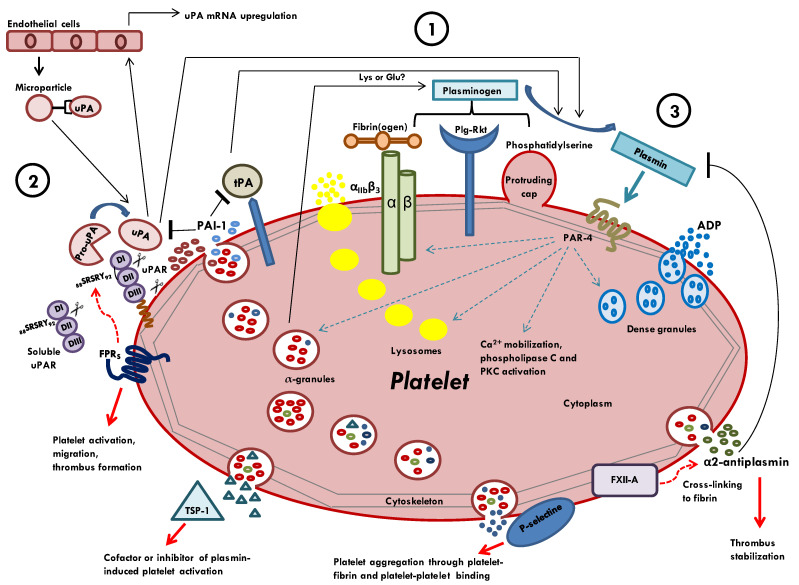
Representation of PAS components contained in or bound to platelets. (**1**) Plasminogen can be released from platelet α-granules after thrombin stimulation and can interact with specific receptors on the platelet surface: αIIbβ3, phosphatidylserine-exposing protruding caps, and/or Plg-Rkt, an integral transmembrane receptor. (**2**) The two main plasminogen activators are tPA and uPA. tPA derives from both platelets and plasma. Platelet receptors for tPA have not yet been identified. uPA can derive from both α-granules and endothelial cell-derived microparticles. uPA bound to platelet surface upregulates uPA synthesis and release by endothelial cells. Platelets express membrane uPA receptor (uPAR) and release GPI-negative form of uPAR (suPAR). Both GPI-positive and GPI-negative uPAR can interact with FPRs, and they could induce platelet activation, migration, and thrombus formation. Platelets are able to release PAI-1, which is an inhibitor of plasminogen activators and α2-antiplasmin. α2-antiplasmin is cross-linked to fibrin by activated coagulation factor XIII (FXIII-A) in order to promote thrombus stabilization. (**3**) Following its activation, plasmin can act as platelet activator. After binding to Protease-Activated Receptor 4 (PAR-4), plasmin induces αIIbβ3 expression; α-, δ-granules and lysosomes release, Ca^2+^ mobilization, phospholipase C and PKC activation, ADP secretion, and P-selectine expression, thus promoting platelet aggregation. Thrombospondin-1 (TSP-1), released from α-granules, could act as cofactor and/or inhibitor of plasmin-induced platelet activation.

## Data Availability

Not applicable.

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
