# Peer review of "Role of Plasminogen Activation System in Platelet Pathophysiology: Emerging Concepts for Translational Applications"

_ijms, 2022, doi:10.3390/ijms23116065_

Round 1

Reviewer 1 Report

This is a well-written and interesting review article that addresses a hot and emerging topic. It is comprehensive and provides a good overview of the different view regarding the role of the platelets in regulating/modulating the fibrinolytic system. I have the following minor comments:

  1. The authors should describe in more detail and highlight the findings that are considered controversial.
  2. It may be better to make the amino acid number listed in the text as "superscript".
  3. There are some very minor typos that need to be fixed. 

Reviewer 2 Report

”The role of plasminogen activation system in platelet pathophysiology: emerging concepts for translational applications" by Napolitano and Montuoris reviewed the less discussed fibrinolytic and antifibrinolytic contributions of platelets and the presence of different members of the PA system in the alpha-granules. It is a very comprehensive and easy-to-read article; I expect it is of interest to the readers. I have two suggestions:

1. It would be good to see a discussion, with examples, on how to apply this knowledge in the clinical setting, as suggested in the title; for example, which target and for pro- or antifibrinolytic treatment.

2. The figure has provided an excellent summary of the article; it is worth drawing the degranulation correctly.
